# Visual Question Answering with Fine-grained Knowledge Unit RAG and Multimodal LLMs

## Abstract

Visual Question Answering (VQA) aims to answer natural language questions based on information present in images. Recent advancements in multimodal large language models (MLLMs) with internalized world knowledge, such as GPT-4o, have demonstrated strong capabilities in addressing VQA tasks. However, in many real-world cases, MLLMs alone are not enough, as they may lack domain-specific or up-to-date knowledge relevant to images and questions. To mitigate this problem, retrieval-augmented generation (RAG) from external knowledge bases (KBs), known as KB-VQA, is promising for VQA. However, effectively retrieving relevant knowledge is not easy. Traditional wisdom typically converts images into text and employs unimodal (*i.e.,* text-based) retrieval, which can lead to the loss of visual information and hinder accurate image-to-image matching. In this paper, we introduce fine-grained knowledge units including both text fragments and entity images, which are extracted from KBs and stored in vector databases. We also designed a knowledge unit retrieval-augmented generation (KU-RAG) method, through fine-grained retrieval and MLLMs. KU-RAG can accurately find corresponding knowledge, and integrate the retrieved knowledge with the internalized MLLM knowledge using a knowledge correction chain for reasoning. Experimental results indicate that our method can significantly enhance the performance of state-of-the-art KB-VQA solutions, with improvements by up to 10%.

## 1 Introduction

**VQA with Knowledge Bases.** Knowledge-based Visual Question Answering (KB-VQA) extends traditional Visual Question Answering (VQA) by incorporating external knowledge to answer questions where image information alone is insufficient (Shah et al., 2019; Marino et al., 2019). However, traditional methods often face limitations in their ability to perform complex reasoning over both visual content and external knowledge sources, as they typically rely on predefined retrieval mechanisms or specific training data (Wu & Mooney, 2022; Yang et al., 2023).

**VQA with MLLMs.** Recently, the emergence of multimodal large language models (MLLMs), such as GPT-4 (Achiam et al., 2023) and LLaVA (Liu et al., 2023a), has introduced new possibilities for VQA. Unlike previous methods, MLLMs serve not only as powerful reasoning engines but also as vast knowledge repositories, with information learned from world knowledge during pretraining (Wang et al., 2024; Liu et al., 2023b). This dual capability enables more nuanced answers. However, the knowledge acquired during training is general and (maybe outdated) world knowledge, limiting the model's ability to respond to domain-specific and update-to-date queries. As shown in Figure 1(a), when using GPT-4 to ask a question about the bridge in the image, it fails to provide an answer due to a lack of relevant knowledge and LLaVA even hallucinated and provided a "false" answer.

**VQA with RAG and MLLMs.** At this point, it becomes necessary to employ KB-VQA approachs, by retrieving information from a database – a process also known as Retrieval-Augmented Generation (RAG) in the context of LLMs (Fan et al., 2024). This typically involves converting images into captions and then performing passage-level retrieval combined with the query. However, this method struggles to handle fine-grained information for question answering, and during the image-to-text modality conversion process, some visual details are inevitably lost. As shown in Figure 1(b), a unimodal, coarse-grained approach fails to retrieve the relevant knowledge.

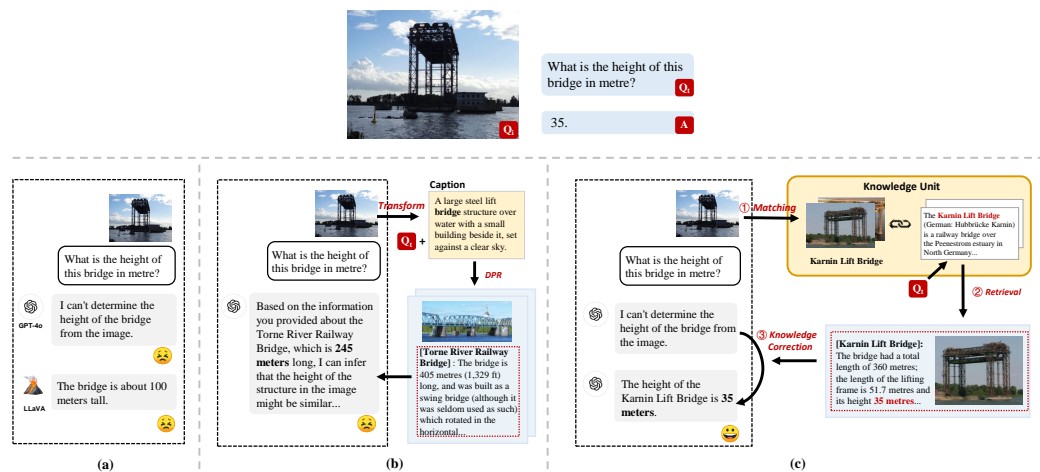

Figure 1: Sample VQA Solutinon with MLLMs: (a) Direct answer without additional knowledge. (b) Single-modality coarse-grained RAG (KB-RAG). (c) Our proposal **KU-RAG**.

**Our Proposal: VQA with Fine-Grained RAG and MLLMs.** Intuitively, in order to accurately find the knowledge corresponding to this bridge, it is necessary to identify the corresponding images through its visual features and then look through the information behind it, as illustrated in Figure 1(c). Following this approach, we propose a "Knowledge Unit" component to bridge the query and specific knowledge. Specifically, we propose a Knowledge Unit Retrieval-Augmented Generation (KU-RAG) method, which is a multimodal, fine-grained, zero-shot retrieval approach covering both data storage and retrieval. As shown in the figure, our method matches the image from the question with images in the database, identifying the relevant knowledge (*e.g.,* "Karnin Lift Bridge") and subsequently retrieving fine-grained knowledge chunks related to it to answer the question more effectively.

Finally, we designed a Knowledge Correction Chain (KCC) to assist in answer generation. The KCC integrates retrieved information into the MLLM's reasoning process and verifies the accuracy of the knowledge generated by MLLMs.

**Contributions.** Our notable contributions are summarized as follows.

- We introduce *knowledge units* that consist of fine-grained multimodal data fragments (*e.g.,* text fragments, entity images, and so on) from external knowledge bases. We store these knowledge units in vector databases, in order to effectively retrieve relevant knowledge for a given visual-based query.

- We propose a knowledge unit retrieval-augmented generation (KU-RAG) method, which retrieves fine-grained knowledge units, employs a knowledge correction chain (KCC) during query inference, and achieves zero-shot for combining retrieved knowledge units with MLLMs.

- The experimental results on multiple different types of KB-VQA benchmarks demonstrate the effectiveness of our method. Our solution effectively improves the performance of MLLM on this task, with enhancements exceeding 10% at best.

## 2 RELATED WORK

### 2.1 KNOWLEDGE-BASED VISUAL QUESTION ANSWERING

Knowledge-based Visual Question Answering (KB-VQA) aims to leverage external knowledge to assist in answering questions about images (Marino et al., 2019; Gardères et al., 2020). In early KB-VQA approaches (Zhu et al., 2020; Gao et al., 2022; Lin & Byrne, 2022), Wikipedia was often used as the external knowledge source for KB-VQA, leading to the common adoption of a retriever-reader framework. This framework first retrieves textual knowledge relevant to the question

and image, and then "reads" the text to predict the answer. However, this passage dense retrieval method is a unimodal, coarse-grained text-to-text retrieval process, which struggles with specific, fine-grained questions such as visual entities (Hu et al., 2023a), events (Yang et al., 2023), and visual information-seeking questions (Chen et al., 2023).

Since the emergence of LLMs, some methods have explored using implicit knowledge from LLMs in addition to retrieving information from databases like Wikipedia. Typically, they convert images into tags or captions and then use GPT to retrieve related knowledge (Gui et al., 2021). However, there is a gap between the query and the LLM's knowledge source. To address this, Hu et al. (2023b) proposed a prompt-guided image captioning method that controls the visual entities in generated captions based on textual queries, replacing general captions with question-dependent ones.

Although some methods attempt to mitigate the loss of visual information by incorporating visual features (Salaberria et al., 2023) or enriching prompts with candidate answers (Shao et al., 2023), they have not completely solved the issue of information loss. In contrast, our approach directly leverages multimodal information for both retrieval and reasoning. In our method, the LLM not only serves as a knowledge base but also plays the role of a reasoner.

## 2.2 MULTIMODAL RETRIEVAL-AUGMENTED GENERATION

Although LLMs possess strong general knowledge answering capabilities, they still face limitations when dealing with domain-specific knowledge, outdated information, and avoiding hallucinations (Gao et al., 2023). To address these issues, Retrieval-Augmented Generation (RAG) was developed. RAG enhances the answering ability of LLMs by retrieving relevant document fragments from external knowledge bases. Specifically, the RAG approach involves multiple modules such as data storage, query optimization, document retrieval, and answer generation. The basic process matches user queries with documents from a large external knowledge base, retrieves relevant document fragments, and generates answers by integrating this information through a generation model. This process is similar to the knowledge retrieval mechanisms used in KB-VQA.

Building on this, RAG has evolved to optimize the retrieval and generation processes. For example, GRAG (Graph Retrieval-Augmented Generation) improves the relevance of information and generation quality by emphasizing the importance of subgraph (Hu et al., 2024). The FiD-RAG (Fusion-in-Decoder RAG) model parallelly fuses multiple retrieved documents during the generation stage, allowing the model to comprehensively integrate background knowledge from different sources (Izacard & Grave, 2020). Moreover, DPR-RAG (Dense Passage Retrieval RAG) introduces dense retrieval techniques that significantly improve retrieval accuracy, quickly locating highly relevant fragments from large document collections (Karpukhin et al., 2020).

Unlike the aforementioned methods, our goal is to build a multimodal, fine-grained RAG model that combines the internal knowledge of LLMs to generate high-quality answers in zero-shot scenarios.

## 3 OUR METHODOLOGY: VQA WITH KNOWLEDGE UNIT RAG AND MLLMS

In this section, we will provide a detailed explanation of the **K**nowledge **U**nit **R**etrieval-**A**ugmented **G**eneration (KU-RAG) method. We construct a novel framework based on KU-RAG for KB-VQA, as illustrated in Figure 2. We will introduce the KB-VQA task in Section 3.2, then present our proposed "knowledge unit" in Section 3.2, followed by a complete overview of KU-RAG in Section 3.3. Finally, we will discuss issues related to KU database management in Section 3.4.

### 3.1 TASK DEFINITION

**Visual Question Answering (VQA):** Given a question $Q$, which consists of an image $Q_i$ and a textual question $Q_t$ related to the content of the image, the task of VQA is to generate an answer $A$ based on the information available in the image and the text. In this setup, the system aims to understand both the visual and textual aspects of the input and provide a relevant response.

**Knowledge-Based Visual Question Answering (KB-VQA):** In KB-VQA, the goal extends beyond the basic VQA task by incorporating external knowledge $K$ stored in knowledge bases (KBs) to answer the question. This external knowledge, which can be categorized as either image knowledge

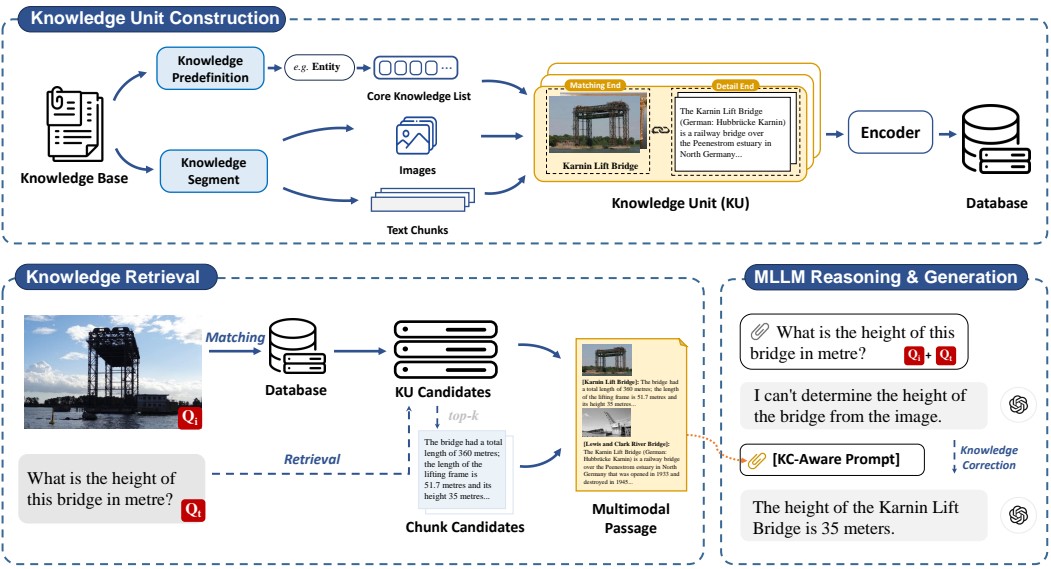

Figure 2: The KU-RAG Framework

$K_i$ or text knowledge $K_t$, is retrieved based on the question $Q$ and is used to generate a more informed and accurate answer $A$.

## 3.2 KNOWLEDGE UNIT

### 3.2.1 DEFINITION

We introduce a new structure called **Knowledge Unit (KU)**. Each KU serves as a knowledge carrier or object generated in combination with the query, such as entities, events, rules, topics, etc., designed to bridge the gap between the query and the database during the actual question-answering process.

For a piece of knowledge, the three most important factors are its image, its name, and detailed textual knowledge. Therefore, we designed each KU as a triplet, consisting of Knowledge Image ($K_i$), Knowledge Name ($K_n$), and Knowledge Text ($K_t$):

$$KU = \{K_i, K_n, K_t\} \tag{1}$$

In the KB-VQA task, image-image or name-name matching is typically used to determine which piece of knowledge a given image belongs to. Hence, we encapsulate $K_i$ and $K_n$ into the **'Matching End'** to link the query and the KU. The purpose of KB-VQA often involves querying the knowledge behind an image, so we refer to $K_t$ as the **'Detail End'**.

### 3.2.2 CONSTRUCTION

**Knowledge Predefinition.** Firstly, we should determine how to extract the knowledge unit with the application scenario and consider the query and database. For example, in an object recognition QA system, different entities can serve as knowledge units; in an event query system, different events can serve as knowledge units; in a corporate rules and regulations query system, a rule-based knowledge unit should be constructed. It is important to note that knowledge units do not necessarily need to be atomic or as fine-grained as possible. Its division should be determined based on the granularity of the data in the query and the database. For instance, a general animal knowledge QA system may only require the general species of an animal (*e.g.,* "cat"), whereas a specific species QA system may require the specific species name (*e.g.,* "Persian cat").

**Knowledge Segment.** Since subsequent steps involve the storage of the knowledge unit, storing textual knowledge $K_t$ within the detail end at the document level, which is a coarse-grained storage method, is highly detrimental to knowledge retrieval (Chen et al., 2021). Additionally, during reason-

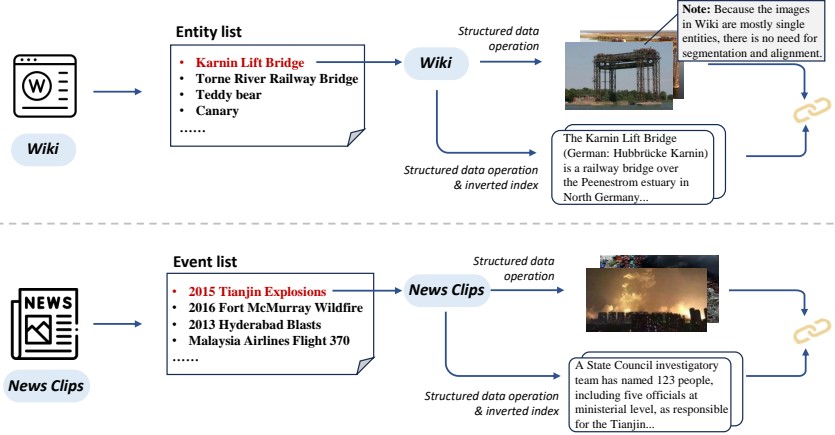

Figure 3: The construction of knowledge units (KU) with entity and event. The entity list comes from the work of Hu et al. (2023a), while the events are sourced from E-VQA (Yang et al., 2023).

ing, this can be limited by the LLM's maximum token capacity, leading to incomplete information. Therefore, we first need to segment the raw data, breaking it down into finer-grained units.

- **Textual data:** Similar to general RAG methods, we next segment all text passage $P = (p_1, p_2, \ldots, p_n)$ in a knowledge base to obtain the smallest retrieval units. Each passage $P$ contains $n$ sentences, *i.e.,* $P_k = (s_1, s_2, \ldots, s_n)$. Considering the importance of knowledge coherence in the KB-VQA task, we adopt a combination of sentence splitting and maximum token limit. In each chunk, as many sentences as possible are retained without exceeding the maximum token limit, and the remaining sentences are assigned to the next chunk. That is, $C = (c_1, c_2, \ldots c_i)$, where $c_j = (s_1, s_2, \ldots s_j)$. $i$ represents the $i$-th chunk, and $j$ represents the $j$-th sentence in the chunk.

- **Image data:** For images pertaining to the same piece of information (such as all images within a news article), we directly extract them, treating each image individually.

**Knowledge Assembly.** After segmenting the text and generating chunks, the next step is to assemble these units with the unprocessed image information to form a knowledge unit. In simple terms, we assemble this multimodal knowledge into knowledge units by leveraging the original structural properties of the knowledge and performing an inverted index on the text. To facilitate understanding, we illustrate this entity-type and event-type knowledge unit shown in Figure 3.

### 3.2.3 STORAGE

Next, we need to store the knowledge contained within the knowledge units. We encode each chunk into a vector using a text encoder and store them in a Faiss database (Douze et al., 2024), which we denote as $D_t$. Considering the need to handle multimodal data in the framework and the possibility of longer text within the chunks, we use Long-CLIP (Zhang et al., 2024) as the vector encoder.

$$V_{c_i} = \text{Encoder}(c_i), D_c = (V_{c1}, V_{c2}, \ldots, V_{cn}) \tag{2}$$

For the images in the knowledge base, we also encode each image using Long-CLIP to obtain visual features and store them in the Meta Faiss vector database, denoted as $D_i$.

$$V_{i_j} = \text{Encoder}(i_j), D_i = (V_{i_1}, V_{i_2}, \ldots, V_{i_n}) \tag{3}$$

### 3.3 KNOWLEDGE UNIT RETRIEVAL FRAMEWORK

In this section, we will introduce our knowledge unit retrieval framework and detail how to achieve knowledge retrieval through knowledge unit and apply it to the KB-VQA task.

As shown in Figure 2, our framework is divided into three modules: **Knowledge Unit Construction**, **Knowledge Retrieval**, and **MLLM Reasoning & Generation**. The knowledge unit construction module mainly transforms raw knowledge into knowledge units and stores them in the database, as illustrate in Section 3.2. The knowledge retrieval module processes the original query, matches it with the corresponding knowledge units, finds the relevant knowledge, and integrates it into a structured, MLLM-readable passage. Finally, combining the original question and the retrieved knowledge, the MLLM Reasoning & Generation module analyzes and generates the answer.

### 3.3.1 QUERY PROCESSING

For the user's input query $Q$, it is first necessary to preprocess and rewrite it to reduce interference during the retrieval process. To find the region in the image related to the question, we propose a query-aware instance segmentation method. Specifically, we first use YOLO (Redmon et al., 2016) to perform instance segmentation on the image, obtaining n segmented instance objects $O = (o_1, o_2, \ldots, o_n)$. We then encode these instances using Long-CLIP, resulting in corresponding vectors $V_o = (v_{o_1}, v_{o_2}, \ldots, v_{o_n})$:

$$V_{o_i} = \text{Encoder}(o_i) \tag{4}$$

Simultaneously, we encode the textual query $Q_t$ into a vector and compute the similarity between it and each vector in $V_o$ to find the object related to the query:

$$V_{q_t} = \text{Encoder}(Q_t), S_n = \text{Sim}(V_{q_t}, V_{o_i}) \tag{5}$$

Here, $S_n = (s_1, s_2, \ldots, s_n)$ represents the similarity values corresponding to each O. We select the object o with the highest similarity that exceeds a threshold $\gamma$ for subsequent retrieval, with its vector denoted as $V_{q_i}$. Of course, sometimes the query may not only be related to one object but also to areas outside the objects or the entire scene in the image. Therefore, if no object meets the criteria or multiple objects meet the criteria, we will encode the entire image and use this encoding for subsequent retrieval:

$$V_{q_i} = \text{Encoder}(Q_i) \tag{6}$$

### 3.3.2 KNOWLEDGE UNIT MATCHING

Next, we use the obtained visual features to match the corresponding knowledge unit. We select the top $k$ knowledge unit items with the highest similarity, denoted as the set $KU'$, where each $ku'$ contains $j$ indices in its detail end.

$$KU = \text{Matching}(V_{q_i}), KU' = \text{top-k}(KU) \tag{7}$$

With $KU' = (ku'_1, ku'_2, \ldots, ku'_k)$. The indices of each $ku'_i$ are represented as $C_i = (c_{i,1}, c_{i,2}, \ldots, c_{i,j})$. Finally, we obtain the combined index set of the knowledge unit:

$$C_{ku} = (C_1, C_2, \ldots, C_n) \tag{8}$$

To integrate KU information into the query while highlighting the importance of certain content words, we rewrite $Q_t$ as: "$Q'_t = Q_t$ [SEP] [KU name] [SEP] keywords", and encode it as:

$$V'_{q_t} = \text{Encode}(Q'_t) \tag{9}$$

Here, "KU name" refers to the name of the matching segment of the retrieved KU, and "keywords" is a list of content words extracted from $Q_t$, separated by commas. "[SEP]" is a special token used to separate different parts. Next, we combine the features of $Q_t$ and calculate the similarity to obtain the top $k$ chunks related to the query, denoted as $C'$:

$$C' = \text{top-k}(\text{Sim}(V_{q_t}, C_{ku}))$$ (10)

with $C' = (c'_1, c'_2, \ldots, c'_k)$.

### 3.3.3 MLLM REASONING AND GENERATION

After retrieving the relevant blocks, the next step is to provide the retrieved information to the MLLM to assist with reasoning and generation. The specific steps are as follows:

**Modal Aligning and Fusing:** First, based on the retrieval results $C'$, we find the corresponding knowledge unit $KU'$ for each chunk and combine its matching end information to form an image with the structure '*[Image][[Name][Chunk Text]]*', where the image corresponding to the $i$-th chunk is denoted as $I'_i$. Notably, if multiple chunks correspond to the same image, to enhance the connection between knowledge and improve processing efficiency, we merge the texts of these chunks into a single image in the format '*[Image][[Name][Chunk $Text_1$][Chunk $Text_2$]...[Chunk $Text_n$]]*'.

**Images Stitching:** Next, we stitch all the images obtained in the previous step to generate a multimodal passage with both image and text information. This multimodal passage $MP$ is as:

$$MP = (I'_1, I'_2, \ldots, I'_n)$$ (11)

**Knowledge Correction Chain:** At this stage, a key challenge is effectively managing the relationships among the 'information in the query', 'the knowledge retrieved', and 'the inherent knowledge of the MLLM', as well as ensuring a fine-grained correspondence between text and images.

In our experiments, we found that when combining the retrieved knowledge with the question for the MLLM to answer, the model tended to prioritize the retrieved information while neglecting its own knowledge. We also attempted to use guiding prompts, such as "*Based on your own knowledge first...*" and "*Focus on the first image and ignore other image...*" to encourage the MLLM to consider its own knowledge before referring to the retrieved information, but the results were unsatisfactory (as demonstrated in Section 4.5).

To address this issue, we design a **Knowledge Correction Chain (KCC)** that guides MLLMs in reasoning through multi-turn dialogue and reading comprehension. In details, we first input the question $Q$ to MLLM to obtain the original answer $A_0$:

$$A_0 = \text{MLLM}(Q)$$ (12)

The purpose of this step is to obtain the pure knowledge of the MLLM regarding the query, without being influenced by the retrieved information mentioned above. Finally, we input the passage $MP$ into the MLLM with a knowledge correction aware (KC-aware) prompt and get the final answer $A$:

> **[KC-aware prompt]:** The initial answer has already been provided. The new image information may either be related or unrelated to the previous input. If this new information conflicts with the initial answer, please update the response accordingly. If no changes are needed, simply output the initial answer again.

$$A = \text{MLLM}(MP, Prompt, (Q, A_0))$$ (13)

In short, the idea of KCC is to shift the MLLM's focus from analyzing the relationship between "information in the query", "the inherent knowledge of the MLLM", and "the knowledge retrieved" to allow "the knowledge retrieved" to correct the MLLM's responses, fostering a reflective process. We have also attempted to use a single prompt to have the MLLM generate and then reflect on its answer, but it still gets influenced by the retrieved information.

In this way, we can fully utilize multimodal information and handle the fine-grained correspondences between them, enhancing the MLLMs' ability to reason and answer questions in complex scenarios.

## 3.4 DATABASE MANAGEMENT

Since our approach operates at the database level, compared to traditional methods, we must also consider issues related to data management. Additionally, there is no need to retrain the entire framework after adding, deleting, or updating data or knowledge.

**Knowledge Addition:** When new knowledge is introduced, we directly chunk the corresponding text. The encoded text vectors are then stored in the existing text vector database $D_c$. Similarly, for images within the knowledge base, we encode them and store the resulting vectors in the image vector database $D_i$.

**Knowledge Unit Management:** After performing operations of the raw data, it is necessary to consider whether corresponding actions need to be taken for the knowledge unit.

1) KU Addition and Update: When new raw data is added, it is essential to assess whether a new KU needs to be introduced. This process primarily involves two steps: first, matching the new knowledge with existing KU. If the similarity of the matching results exceeds the threshold $\alpha$, the index containing the new keywords is added to the corresponding KU through keyword matching. If no matching result exceeds the threshold, a new KU is constructed according to the KU construction rules and the new keywords present in the chunk.

2) KU Deletion: After deleting a chunk from the raw data, it is necessary to check whether the related KU is still valid to reduce storage usage. Specifically, after deleting the chunk indexed as $i$, all KUs containing this index should be checked. If a certain KU has an empty detail end (*i.e.,* no remaining values), that KU can be deleted.

## 4 EXPERIMENT

### 4.1 DATASET

To validate the effectiveness of our method, we selected four representative KB-VQA datasets, each with its own focus areas:

- **OVEN** (Hu et al., 2023a): An Open-domain Visual Entity Recognition dataset, primarily examining the ability to recognize the names of visual entities.
- **INFO SEEK** (Chen et al., 2023): An extension of the OVEN dataset, focusing on the coarse-grained knowledge behind entities, environments, etc., in images. It requires identifying the image and then discovering the knowledge behind it.
- **OK-VQA** (Marino et al., 2019): A classic KB-VQA dataset focusing on open-domain knowledge, featuring images paired with open-ended questions.
- **E-VQA** (Yang et al., 2023): An event-centric dataset, primarily evaluating the ability to recognize events and the knowledge behind them.

Table 1 shows some characteristics of each dataset. The more stars in question granularity, the finer the question. The higher the popularity of knowledge, the more general it is, meaning the MLLM is more likely to have learned it during pre-training. Note that since our method is conducted in a zero-shot setting, we only selected the test sets of these datasets. Due to the large size of the original test sets for OVEN and INFO SEEK, we sampled some examples using an arithmetic sequence for testing, and the term '$s$' is used in Table 1 and subsequent experimental results to indicate this.

### 4.2 BASELINE

Due to variations in the formats and objectives of each dataset, there is no single unified state-of-the-art (SOTA) model across all of them. To ensure a fair comparison, we select the best-performing model for each dataset as its respective SOTA baseline. Specifically:

- For **OVEN** dataset, we use PaLI-17B (Chen et al., 2022), as reported by Hu et al. (2023a).
- For **INFO SEEK** and **OK-VQA** datasets, the SOTA model is PaLI-X (Chen et al., 2022), as reported in the work of Chen et al. (2023).

Table 1: Characteristics of different dataset.

| Dataset | Tests Number | Knowledge Source | Knowledge Granularity | Knowledge Popularity |
|---|---|---|---|---|
| OVEN$_s$ | 23,650 | Wiki | ★★ | ★★ |
| INFO SEEK$_s$ | 11,600 | Wiki | ★★★ | ★★ |
| OK-VQA | 5,064 | Wiki | ★★ | ★★★ |
| E-VQA | 1,819 | News | ★★★ | ★ |

Table 2: Main results of the experiment. And $^\dagger$ indicates that the result is from experiments conducted on the full version of the test set, sourced respectively from Hu et al. (2023a) and Chen et al. (2023).

| Model | Dataset | | | |
|---|---|---|---|---|
| | OVEN$_s$ | INFO SEEK$_s$ | OK-VQA | E-VQA |
| SOTA (trained) | 21.70$^\dagger$ | 22.10$^\dagger$ | 66.10 | 19.42 |
| gpt-4o (zero-shot) | 22.30 | 36.05 | 75.52 | 15.17 |
| gpt-4o + KU-RAG (zero-shot) | **26.50** | **38.35** | **77.23** | **26.16** |

- For **E-VQA** dataset, we adopt the best results of the SOTA model MuKEA (Ding et al., 2022), as reported Yang et al. (2023).

## 4.3 EXPERIMENTAL ENVIRONMENT

Our experiments were conducted on two RTX 4090 GPUs using the base version API of GPT-4o as the MLLM. In our methods' settings, OVEN, INFO SEEK, and OK-VQA all use entities as the knowledge unit, while E-VQA uses events as the knowledge unit. For the recall of knowledge units and chunks, the top-k is set to 3. For the experiment evaluation, we used accuracy as the metric.

## 4.4 MAIN RESULT

As shown in Table 2, we have the following findings.

**Zero-shot Capability of MLLM:** MLLM demonstrates remarkable zero-shot capability, especially in image understanding and reasoning. Compared to the previous SOTA model for KB-VQA, MLLM shows improvements of 0.6%, 13.95%, and 9.42% on the OVEN, INFO SEEK, and OK-VQA datasets, respectively. This is mainly due to the extensive world knowledge accumulated during MLLM's pre-training phase. However, since the E-VQA dataset involves less popular news knowledge, MLLM's performance in this area is not as strong as that of the specially trained SOTA model.

**Superior Performance of MLLM+KU-RAG:** Our method, MLLM+KU-RAG, performs excellently across all datasets. In a zero-shot scenario, even without reviewing the training set knowledge, MLLM+KU-RAG outperforms the existing SOTA models by 4.8%, 16.25%, 11.13%, and 6.74% on the four datasets, respectively, validating the powerful performance of our approach.

**Enhancement of MLLM by KU-RAG:** Combining KU-RAG with gpt-4o results in performance improvements of 4.2%, 2.3%, 1.7%, and 10.99% on the respective tasks. The largest improvement is seen in the E-VQA dataset, as it involves less popular knowledge, and the new knowledge provided by KU-RAG significantly enhances model performance. In contrast, the improvement on the OK-VQA dataset is smaller because it involves open-domain general knowledge, which MLLM may have already encountered during pre-training, allowing it to answer effectively.

## 4.5 ABLATION RESULT

To validate the effectiveness of each component in our proposed method, we designed ablation experiments comparing the following models:

- **w/o KCC:** This model omits the knowledge correction chain (KCC), relying instead on the model's analysis of the question and the retrieved information in a single-turn Q&A setup.

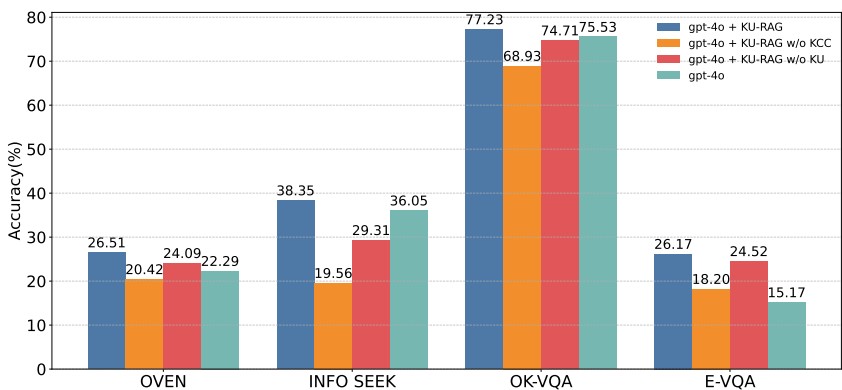

Figure 4: The Results of Ablation Study

- **w/o UK:** This model removes the fine-grained retrieval approach (*i.e.,* knowledge unit), converting the information from images into captions and using a text-only retrieval modality.

Additionally, we included the full implementation of the MLLM+KU-RAG method, as well as a standalone MLLM method. The experimental results are shown in Figure 4. From the figure, we can draw the following conclusions:

**Effectiveness of MLLM+KU-RAG:** The MLLM+KU-RAG method consistently achieves the highest performance, which demonstrates that every part of this method is indispensable.

**Impact of Removing KCC:** Removing KCC and using single-turn dialogue markedly reduces model performance across four datasets, with decreases of 6%, 18.79%, 8.3%, and 7.97%, respectively. Except for the E-VQA dataset, the model's performance is inferior to using only gpt-4o. This likely occurs because the model struggles to effectively focus on the original question's image and manage the logical relationships between the query information, its own knowledge, and the retrieved knowledge. Consequently, some questions, which the model could originally answer correctly, are answered incorrectly due to interference from the injected information.

**Impact of Removing UK:** Removing UK and adopting a coarse-grained, single-modality retrieval approach results in a slight performance drop across datasets, with the most significant decrease observed in the INFO SEEK dataset (9.04%). This is partly because INFO SEEK requires matching detailed image content and background knowledge, and converting the original image to captions loses a substantial amount of visual information. As illustrated by examples in Figure 1, it's challenging to accurately match "Karnin Lift Bridge" using just the text "bridge," let alone find corresponding background knowledge. Furthermore, introducing incorrect knowledge adds noise, impeding the MLLM's reasoning process and leading to erroneous results. The smallest performance drop is observed in the E-VQA dataset (1.65%), likely because, in this dataset, the images primarily serve to supplement information, allowing text-only retrieval to still achieve reasonably good matches.

## 5 CONCLUSION

In this paper, we introduce the Knowledge Unit Retrieval-Augmented Generation (KU-RAG) method, aimed at enhancing MLLMs by incorporating fine-grained retrieval of domain-specific knowledge. To improve the effectiveness of retrieval, we propose the concept of "knowledge units", which allows for more targeted access to relevant information. Furthermore, we design a knowledge correction chain strategy to verify and refine the retrieved knowledge, which can mitigate errors and hallucinations, enhancing the overall reliability and coherence of the generated answers in VQA tasks. Our experimental results demonstrate significant performance gains across multiple KB-VQA benchmarks, highlighting the effectiveness of our approach. Future research directions could explore dynamic knowledge updates stategy to improve multimodal retrieval and semantic integration.

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
