# OpenReview forum: "Visual Question Answering with Fine-grained Knowledge Unit RAG and Multimodal LLMs"
_ICLR.cc/2025/Conference — ICLR 2025 Conference Withdrawn Submission_

### Official Review · Reviewer_2kUB · 2024-10-27

**Soundness:** 2
**Presentation:** 1
**Contribution:** 2
**Rating:** 3
**Confidence:** 5

**Summary:**

This paper leverages knowledge base and proposes retrieval-augmented generation to enhance the input information for VQA tasks.

**Strengths:**

1-This research direction / perspective is interesting and provides the community with some insights.

2-The figures are well-presented.

**Weaknesses:**

1-The writting can be improved: especially Sec.1. Current Sec. 1 is more like Related work instead of Introduction. The method is easy to understand, but the writing in Sec.3 makes me confusing.

2-Insufficient experiments: only conduct exp with GPT-4o. Should combine proposed KU-RAG with both open-sourced and proprietary MLLMs, and at least 3-5 different MLLMs.

3-The method lacks novelty: this method is more like a combination of multiple engineering techniques. Extracting from KB is widely used in KVQA tasks, knowledge correction chain is CoT.

**Questions:**

The weakness is in the above part.

Strongly suggests the authors improve both method and writing.

---

### Official Review · Reviewer_YnZq · 2024-10-29

**Soundness:** 2
**Presentation:** 3
**Contribution:** 2
**Rating:** 3
**Confidence:** 4

**Summary:**

This paper introduced knowledge units with fine-grained multimodal data fragments for knowledge-based VQA. The authors further proposed a knowledge unit retrieval-augmented generation (KU-RAG) method with a knowledge correction chain for zero shot KB-VQA by combining retrieved knowledge units with MLLMs. Experiments on GPT-4o validate the effectiveness of the proposed knowledge base and the method. However, more experiments and analysis should be conducted to validate the effectiveness of the proposed knowledge base and method.

**Strengths:**

1.	This paper proposed fine-grained knowledge units for KB-VQA, which can boost model performance for KB-VQA.
2.	This paper introduced the Knowledge Correction Chain (KCC) that guides MLLMs in reasoning through multi-turn dialogue and reading comprehension.

**Weaknesses:**

1.	The details of the constructed knowledge base are not provided (e.g., knowledge source, scale, length, etc.).
2.	Experiments are limited on GPT-4o, experiments on more open-source MLLMs (e.g., LLaVA) should be included.
3.	To validate the effectiveness of the proposed knowledge base, experiments should be conducted on the comparison of the proposed knowledge base with other knowledge bases.
4.	The proposed framework is similar to the traditional multimodal RAG [1], which leads to low novelty.

[1] Lin W, Chen J, Mei J, et al. Fine-grained late-interaction multi-modal retrieval for retrieval augmented visual question answering[J]. Advances in Neural Information Processing Systems, 2023, 36: 22820-22840.

**Questions:**

See Weaknesses.

---

### Official Review · Reviewer_GAdV · 2024-10-31

**Soundness:** 2
**Presentation:** 3
**Contribution:** 3
**Rating:** 5
**Confidence:** 4

**Summary:**

This paper introduces an RAG framework for enhancing MLLMs in VQA. The authors build knowledge units as outside knowledge sources and propose a KU-RAG framework to retrieve relevant text to assist MLLM in answering the visual question. The knowledge correction chain
is designed to improve reasoning by correcting mistake answers with a prompt-based approach.

**Strengths:**

- The research problem addressed in this work is both important and interesting, aiming to enhance Multimodal Large Language Models (MLLMs) with Retrieval-Augmented Generation (RAG).
-  This paper is well-illustrated and easy to understand.
-  The paper proposes a useful framework for enhancing MLLMs using RAG.

**Weaknesses:**

- Unclear Contribution of Knowledge Units: The knowledge units represent a straightforward implementation for organizing the existing knowledge base with both images and text. While I understand that the knowledge unit serves as the foundation for the proposed RAG framework, the technical contribution in this area is somewhat limited.
- Necessity of KCC: The Knowledge Correction Component (KCC) is designed to replace incorrect retrieval-augmented answers with directly generated answers from the inherent knowledge of MLLMs. However, the primary purpose of RAG is to equip MLLMs with external knowledge. If I understand correctly, KCC merely remedies poor retrieval results and relies on the MLLM's ability to distinguish incorrect results. If the authors could demonstrate the effectiveness of KCC in weaker MLLMs, such as LLaVA1.5[1], its application would be more practical.
- Lack of Baselines: My major concern in the experiments section is the lack of competitive baselines. Although the authors compare with trained and text-retrieval baselines, no baseline is presented for multimodal RAG. The zero-shot GPT-4o baseline already outperforms the "SOTA (trained)" methods in Table 2. If the authors could provide comparisons with the following RAG strategies:
1. CLIP image-to-image retrieval (EchoSight [2])
2. CLIP text-to-image retrieval (InfoSeek [3])

the experimental results would be more persuasive.

- Limited Choice of MLLMs: The authors only conduct experiments with the powerful GPT-4o, raising concerns that the proposed methods may only work with costly, large MLLMs. Could the authors provide results using popular open-source MLLMs such as LLaVA1.5 or LLaVA-Next?
References:
[1] Improved Baselines with Visual Instruction Tuning
[2] EchoSight: Advancing Visual-Language Models with Wiki Knowledge
[3]Can Pre-trained Vision and Language Models Answer Visual Information-Seeking Questions?

**Questions:**

Please kindly answer the question in the weakness section.

---

### Official Review · Reviewer_dKH5 · 2024-11-04

**Soundness:** 3
**Presentation:** 2
**Contribution:** 2
**Rating:** 3
**Confidence:** 4

**Summary:**

The paper proposed a knowledge-unit RAG to enhance knowledge-based vqa task. The authors explained in detail the knowledge unit construction process, KU retrieval, then the visual question answering with a MLLM. A key component is the Knowledge Correction Chain (KCC) for answer verification and correction. The KCC is designed to integrate retrieved external knowledge with MLLM's internal knowledge for answer generation. Obvious improvements are reported on several KB-VQA benchmarks.

**Strengths:**

1. Introducing the Knowledge Correction Chain into VQA task is an interesting attempt, which should help alleviating hallucination of MLLMs.
2. The proposed knowledge unit construction can be adapted to other multimodality tasks that requires the assistance of external knowledge.

**Weaknesses:**

1. Some phrases like "Traditional wisdom" flags the potential usage of LLMs in composing the draft.
2. The design of KCC is still a naive experimental attempt which may not cast a positive impact on real knowledge-based vqa tasks. Despite the promising benchmarking results, the nature of KCC means that error propagation and knowledge conflicts are inevitable when applied to real-world vqa cases.

**Questions:**

1. Can you explain more on the necessity of constructing knowledge units? Why should it surpasses querying multimodal knowledge individually?
2. Does knowledge units construction need to be conducted for each individual dataset or shared knowledge space is possible?
3. How did you measure the quality of knowledge units constructed?

---

### Note · Authors · 2024-11-13

I have read and agree with the venue's withdrawal policy on behalf of myself and my co-authors.